# Rates of Convergence for Large-scale Nearest Neighbor Classification

**Xingye Qiao**
Department of Mathematical Sciences
Binghamton University
New York, USA
qiao@math.binghamton.edu

**Jiexin Duan**
Department of Statistics
Purdue University
West Lafayette, Indiana, USA
duan32@purdue.edu

**Guang Cheng**
Department of Statistics
Purdue University
West Lafayette, Indiana, USA
chengg@purdue.edu

## Abstract

Nearest neighbor is a popular class of classification methods with many desirable properties. For a large data set which cannot be loaded into the memory of a single machine due to computation, communication, privacy, or ownership limitations, we consider the divide and conquer scheme: the entire data set is divided into small subsamples, on which nearest neighbor predictions are made, and then a final decision is reached by aggregating the predictions on subsamples by majority voting. We name this method the big Nearest Neighbor (bigNN) classifier, and provide its rates of convergence under minimal assumptions, in terms of both the excess risk and the classification instability, which are proven to be the same rates as the oracle nearest neighbor classifier and cannot be improved. To significantly reduce the prediction time that is required for achieving the optimal rate, we also consider the pre-training acceleration technique applied to the bigNN method, with proven convergence rate. We find that in the distributed setting, the optimal choice of the neighbor $k$ should scale with both the total sample size and the number of partitions, and there is a theoretical upper limit for the latter. Numerical studies have verified the theoretical findings.

## 1 Introduction

In this article, we study binary classification for large-scale data sets. Nearest neighbor (NN) is a very popular class of classification methods. The $k$NN method searches for the $k$ nearest neighbors of a query point $x$ and classify it to the majority class among the $k$ neighbors. NN methods do not require sophisticated training that involves optimization but are memory-based (all the data are loaded into the memory when predictions are made.) In the era of big data, the volume of data is growing at an unprecedented rate. Yet, the computing power is limited by space and time and it may not keep pace with the growth of the data volume. For many applications, one of the main challenges of NN is that it is impossible to load the data into the memory of a single machine [36]. In addition to the memory limitation, there are other important concerns. For example, if the data are collected and stored at several distant locations, it is challenging to transmit the data to a central location. Moreover, privacy and ownership issues may prohibit sharing local data with other locations. Therefore, a distributed approach which avoids transmitting or sharing the local raw data is appealing.

There are new algorithms which are designed to allow analyzing data in a distributed manner. For example, [36] proposed distributed algorithms to approximate the nearest neighbors in a scalable fashion. However, their approach requires subtle and careful design of the algorithm which is not generally accessible to most users who may work on many different platforms. In light of this, a more general, simple and user-friendly approach is the divide-and-conquer strategy. Assume that the total sample size in the training data is $N$. We divide the data set into a number of $s$ subsets (or the data are already collected from $s$ locations and stored in $s$ machines to begin with.) For simplicity, we assume that each subset size is $n$ so that $N = sn$. We may allow the number of subsets $s$ to grow with the total sample size $N$, in the fashion of $s = N^\gamma$. For each query point, $k$NN classification is conducted on each subset, and these local predictions are pooled together by majority voting. Moreover, since the $k$NN predictions are made at local locations, and only the predictions (instead of the raw data) are transmitted to the central location, this approach is more efficient in terms of communication cost and it reduces (if not eliminates) much of the privacy and ownership concerns. The resulting classifier, which we coin as the big Nearest Neighbor (bigNN) classifier, can be easily implemented by a variety of users on different platforms with minimal efforts. We point out that this is not a new idea, as it is essentially an ensemble classifier using the majority vote combiner [10]. Moreover, a few recent work on distributed regression and principal component analysis follow this direction [46, 11, 5, 32, 18, 38, 47]. However, distributed classification results are much less developed, especially in terms of the statistical performance of the ensemble learner. Our approach is different from bagging (or bootstrap in general) [7], another type of ensemble estimator. Bagging was historically proposed to enhance the prediction accuracy by reducing variance and conduct statistical inference even when the sample size is not large enough. They are not our concern here. Our algorithm is motivated by the need to maintain data decentralisation/privacy and enhance speed performance, when the sample size is too large.

In practice, NN methods are often implemented by algorithms capable of dealing with large-scale data sets, such as the kd-tree [6] or random partition trees [14]. However, even these methods have limitations. As the definition of "large-scale" evolves, it would be good to know if the divide-and-conquer scheme described above may be used in conjunction to these methods. There are related methods such as those that rely on efficient $k$NNs or approximate nearest neighbor (ANN) methods [2, 1, 26, 39]. However, little theoretical understanding in terms of the classification accuracy of these approximate methods has been obtained (with rare exceptions like [21].)

Some quantization strategies [30, 28] have been proposed recently to scale up NNs to large datasets. They often start with a $r$-net, which is a collection of data points that quantize the training data. The average response value or the majority class label of those training points that fall into each cell is then assigned to these cells. This is quite similar to the denoising scheme in Xue and Kpotufe [44], in which quantization is achieved by random subsamplings. However, all these quantization schemes have heavy computational burden in terms of preprocessing: for a very large training data, assigning the weights for each cell will be as difficult as predicting the class label of a query point using kNN. From this perspective, we propose a denoised bigNN algorithm to shorten the preprocessing time of quantization-based approaches without sacrificing the accuracy.

The asymptotic consistency and convergence rates of the NN classification have been studied in details. See, for example, Fix and Hodges Jr [19], Cover and Hart [12], Devroye et al. [15], Chaudhuri and Dasgupta [9]. However, there has been little theoretical understanding about the bigNN classification. In particular, one may ask whether the bigNN classification performs as well as the oracle NN method, that is, the NN method applied to the entire data set (which is difficult in practice due to the aforementioned constraints and limitations, hence the name "oracle".) To our knowledge, our work is the first one to address the classification accuracy of the ensemble NN method, from a statistical learning theory point of view, and build its relation to that of its oracle counterpart. Much progress has been made for the latter. Cover [13], Wagner [43], and Fritz [20] provided distribution-free convergence rates for NN classifiers. Later works [31, 23] gave rate of convergence in terms of the smoothness of class conditional probability $\eta$, such as Hölder's condition. Recently, Chaudhuri and Dasgupta [9] studied the convergence rate under a condition more general than the Hölder's condition of $\eta$. In particular, a smooth measure was proposed which measures the change in $\eta$ with respect to probability mass rather than distance. More recently, Kontorovich and Weiss [29] proposed a strongly Bayes consistent margin-regularized 1-NN; Kontorovich et al. [28] proved a sample-compressed 1-NN based multiclass learning algorithm is Bayes consistent. In addition, under the so-called margin condition, and assumptions on the density function of the covariates, Audibert and Tsybakov [3]

showed a faster rate of convergence. See Kohler and Krzyzak [27] for results without assuming the density exists. Some other related works about NN methods include Hall et al. [24], which gave an asymptotic regret formula in terms of the number of neighbors, and Samworth [37], which gave a similar formula in terms of the weights for a weighted nearest neighbor classifier. Sun et al. [40] took the stability measure into consideration and proposed a classification instability (CIS) measure. They gave an asymptotic formula of the CIS for the weighted NN classifier.

In this article, we give the convergence rate of the bigNN method under the smoothness condition for $\eta$ established by Chaudhuri and Dasgupta [9], and the margin condition. It turns out that this rate is the same rate as the oracle NN method. That is, by divide and conquer, *one does not lose the classification accuracy in terms of the convergence rate*. We show that the rate has a minimax property. That is, with some density assumptions, this rate cannot be improved. We find out that the optimal choice of the number of neighbors $k$ must scale with the overall sample size and number of splits, and there is an upper limits on how much splits one may use. To further shorten the prediction time, we study the use of the denoising technique [44], which allows significant reduction of the prediction time at a negligible loss in the accuracy under certain conditions, which are related to the upper bound on the number of splits. Lastly, we verify the results using an extensive simulation study. As a side product, we also show that the convergence rate of the CIS for the bigNN method is also the same as the convergence for the oracle NN method, which is a sharp rate previously proven. All these theoretical results hold as long as the number of divisions does not grow too fast, i.e., slower than some rate determined by the smoothness of $\eta$.

## 2 Background and key assumptions

Let $(\mathcal{X}, \rho)$ be a separable metric space. For any $x \in \mathcal{X}$, let $B^o(x, r)$ and $B(x, r)$ be the open and closed balls respectively of radius $r$ centered at $x$. Let $\mu$ be a Borel regular probability measure on $(\mathcal{X}, \rho)$ from which $X$ are drawn. We focus on binary classification in which $Y \in \{0, 1\}$; given $X = x$, $Y$ is distributed according to the class conditional probability function (also known as the regression function) $\eta : \mathcal{X} \mapsto \{0, 1\}$, defined as $\eta(x) = \mathbb{P}(Y = 1 | X = x)$, where $\mathbb{P}$ is with respect to the joint distribution of $(X, Y)$.

**Bayes classifier, regret, and classification instability**

For any classifier $\tilde{g} : \mathcal{X} \mapsto \{0, 1\}$, the risk $\tilde{g}$ is $R = \mathbb{P}(\tilde{g}(X) \neq Y)$. The Bayes classifier, defined as $g(x) = \mathbb{1}\{\eta(x) > 1/2\}$, has the smallest risk among all measurable classifier. The risk for the Bayes classifier is denoted as $R^* = \mathbb{P}(g(X) \neq Y)$.

The excess risk of classifier $\tilde{g}$ compared to the Bayes classifier is $R - R^*$, which is also called the regret of $\tilde{g}$. Note that since the classifier $\tilde{g}$ is often driven by a training data set that is by itself random, both the regret and the risk are random quantities. Hence, sometimes we may be interested in the expected value of the risk $\mathbb{E}_N R$, where the expectation $\mathbb{E}_N$ is with respect to the distribution of the training data $\mathcal{D}$.

Sometimes, we call the algorithm that maps a training data set $\mathcal{D}$ to the classifier function $\tilde{g} : \mathcal{X} \mapsto \{0, 1\}$ a "classifier". In this sense, classification instability (CIS) was proposed to measure how sensitive a classifier is to sampling of the data. In particular, the CIS of a classifier is defined as

$$\text{CIS} \equiv \mathbb{E}_{\mathcal{D}_1, \mathcal{D}_2}[\mathbb{P}_X(\phi_1(X) \neq \phi_2(X) | \mathcal{D}_1, \mathcal{D}_2)],$$

where $\phi_1$ and $\phi_2$ are the classification functions trained based on $\mathcal{D}_1$ and $\mathcal{D}_2$, which are independent copies of the training data [40].

**Big Nearest Neighbor classifiers**

In practice, we have a large training data set $\mathcal{D} = \{(X_i, Y_i), i = 1, \ldots, N\}$, and it may be evenly divided to $s = N^\gamma$ subsamples with $n = N^{(1-\gamma)}$ observations in each. For any query point $x \in \mathcal{X}$, its $k$ nearest neighbors in the $j$th subsample are founded, and the average of their class labels $Y_i$ is denoted as $\hat{Y}^{(j)}(x)$. Denote $g_{n,k}^{(j)}(x) = \mathbb{1}\{\hat{Y}^{(j)}(x) > 1/2\}$ as the $j$th binary $k$NN classifier based on the $j$th subset. Finally, a majority voting scheme is carried out so that the final bigNN classifier is $g_{n,k,s}^*(x) = \mathbb{1}\{\frac{1}{s} \sum_{j=1}^{s} g_{n,k}^{(j)}(x) > 1/2\}$. In this article, we are interested in the risk of $g_{n,k,s}^*$, denoted as $R_{n,k,s}^*$, its corresponding regret, and its CIS.

**Key assumptions**

Many results in this article rely on the following commonly used assumptions. The $(\alpha, L)$-smoothness assumption ensures the smoothness of the regression function $\eta$. In particular, $\eta$ is $(\alpha, L)$-smooth if for all $x, x' \in \mathcal{X}$, there exist $\alpha, L > 0$, such that

$$|\eta(x) - \eta(x')| \leq L\mu(B^o(x, \rho(x, x')))^\alpha.$$

Chaudhuri and Dasgupta [9] pointed out that this is more general than, and is closely related to the Hölder's condition when $\mathcal{X} = \mathbb{R}^d$ ($d$ is the dimension), which states that $\eta$ is $\alpha_H$-Hölder continuous if there exist $\alpha_H, L > 0$ such that $|\eta(x) - \eta(x')| \leq L\|x - x'\|^{\alpha_H}$. Moreover, Hölder's continuity implies $(\alpha, L)$-smoothness, with the equality

$$\alpha = \alpha_H \cdot d^{-1}. \tag{1}$$

This transition formula will be useful in comparing our theoretical results with the existing ones that are based on the Hölder's condition.

The second assumption is the popular margin condition [35, 41, 3]. The joint distribution of $(X, Y)$ satisfies the $\beta$-margin condition if there exists $C > 0$ such that

$$\mathbb{P}(|\eta(X) - 1/2| \leq t) \leq Ct^\beta, \quad \forall t > 0.$$

## 3 Main results

Our first main theorem concerns the regret of the bigNN classifier.

**Theorem 1.** *Set $k = k_o n^{2\alpha/(2\alpha+1)} s^{-1/(2\alpha+1)} \to \infty$ as $N \to \infty$ where $k_o$ is a constant. Under the $(\alpha, L)$-smoothness assumption of $\eta$ and the $\beta$-margin condition, we have*

$$\mathbb{E}_n R_{n,k,s} - R^* \leq C_0 N^{-\alpha(\beta+1)/(2\alpha+1)}.$$

The rate of convergence here appears to be independent of the dimension $d$. However, the theorem can be stated in terms of the Hölder's condition instead, which leads to the rate $N^{-\alpha_H(\beta+1)/(2\alpha_H+d)}$, due to equality (1), which now depends on $d$. It would be insightful to compare the bound derived here for the bigNN classifier with the oracle NN classifier. Theorem 7 in [9] showed that under almost the same assumptions, with a scaled choice of $k$ in the oracle $k$NN method, the convergence rate for the oracle method is also $N^{-\alpha(1+\beta)/(2\alpha+1)}$. This means that divide and conquer does not compromise the rate of convergence of the regret when it is used on the $k$NN classifier.

As a matter of fact, the best known rate among nonparametric classification methods under the margin assumption was $N^{-\alpha_H(1+\beta)/(2\alpha_H+d)}$ (Theorems 3.3 and 3.5 in [3]), which, according to (1), was the same rate for bigNN here and for oracle NN derived in [9]. In other words, the rate we have is sharp. It was proved that the optimal weighted nearest neighbor classifiers (OWNN) [37], bagged nearest neighbor classifiers [25] and the stabilized nearest neighbor classifier [40] can achieve this rate. See Theorem 2 of the supplementary materials of Samworth [37] and Theorem 5 of Sun et al. [40].

Our next theorem concerns the CIS of the bigNN classifier.

**Theorem 2.** *Set the same $k$ as in Theorem 1 ($k = k_o n^{2\alpha/(2\alpha+1)} s^{-1/(2\alpha+1)}$). Under the $(\alpha, L)$-smoothness assumption and the $\beta$-margin condition, we have*

$$CIS(bigNN) \leq C_0 N^{-\alpha\beta/(2\alpha+1)}.$$

Again, we remark that the best known rate for CIS for a non-parameter classification method (oracle $k$NN included) is $N^{-\alpha_H\beta/(2\alpha_H+d)}$ (Theorem 5 in [40]), where $\alpha_H$ is the power parameter in the Hölder's condition. This is exactly the rate we derived here for bigNN classifier by noting (1).

We remark that the optimal number of neighbors for the oracle $k$NN is at the order of $N^{2\alpha/(2\alpha+1)}$, while the optimal number of neighbors for each local classifier in bigNN is at the order of $n^{2\alpha/(2\alpha+1)} s^{-1/(2\alpha+1)}$ which is not equal to $n^{2\alpha/(2\alpha+1)}$ (that is, the optimal choice of $k$ for the oracle above with $N$ replaced by $n$.) In other words, the best choice of $k$ in bigNN will lead to suboptimal performance for each local classifier. However, due to the aggregation via majority voting, these suboptimal local classifiers will actually ensemble an optimal bigNN classifier.

Moreover, $k = k_o n^{2\alpha/(2\alpha+1)} s^{-1/(2\alpha+1)}$ should grow as $N$ grows. In view of the facts that $s = N^\gamma$ and $n = N^{1-\gamma}$, this implies an upper bound on $s$. In particular, $s$ should be less than $N^{2\alpha/(2\alpha+1)}$. Conceptually, there exist notions of bias due to small sample size and bias/variance trade-off for ensembles. If $s$ is too large and $n$ too small, then the 'bias' of the base classifier on each subsample tends to increase, which can not be averaged away by the $s$ subsamples.

Lastly, bigNN may be seen as comparable to the bagged NN method [25]. In that context, the sampling fraction is $n/N = 1/s = N^{-\gamma} \to 0$. [25] suggested that when sampling fractions converge to 0, but the resample sizes diverges to infinity, the bagged NN converges to the Bayes rule. Our work gives a convergence rate in addition to the fact that the regret will vanish as $N$ grows.

# 4    Pre-training acceleration by denoising

While the oracle $k$NN method or the bigNN method can achieve significantly better performance than 1-NN when $k$ and $s$ are chosen properly, in practice, many of the commercial tools for nearest neighbor search are optimized for 1-NN only. It is known that for statistical consistency, $k$ should grow as $N$ to infinity. This imposes practical challenge for the oracle $k$NN to search for the $k$ nearest neighbors from the training data, in which $k$ could potentially be a very large number. Even in a bigNN in which $k$ at each subsample is set to be 1, to achieve statistical consistency one requires $s$ to grow with $N$. These naturally lead to the practical difficulty that the prediction time is very large for growing $k$ or $s$ in the presence of large-scale data sets. In Xue and Kpotufe [44], the authors proposed a denoising technique to shift the time complexity from the prediction stage to the training stage. In a nutshell, denoising means to pre-train the data points in the training set by re-labelling each data point by its global $k$NN prediction (for a given $k$). After each data point is pre-trained, at the time of prediction, the nearest neighbors of the query point from among a small number (say $I$) of subsamples of the training data are identified, and the majority class among these 1-NNs becomes the final prediction for the query point. Note that under this acceleration scheme, at the prediction stage, one only needs to conduct 1-NN search for $I$ times, and each time from a subsample with size $m \ll N$; hence the prediction time is significantly reduced to almost the same as the 1-NN. Xue and Kpotufe [44] further proved that at some vanishing subsampling ratio, the denoised NN can achieve the prediction accuracy at the same order as that of the $k$NN. Note that denoising does not work by ignoring a lot of data all together, but by extraditing the information ahead of time (during preprocessing) and not bothering with the entire data later on at the time of the prediction.

In this section we consider using the same technique to accelerate the prediction time for large data sets in conjunction with the bigNN method. The pre-training step in Xue and Kpotufe [44] was based on, by default, the oracle $k$NN, which is not realistic for very large data sets. We consider using the bigNN to conduct the pre-training, followed by the same 1-NN searches at the prediction stage. Our work and the work of [44] can be viewed as supplementary to each other. As [44] shifted the computational time from the prediction stage to the training stage, we reduce the computational burden at the training stage by using bigNN instead of the oracle NN. A subtlety is that the denoising algorithm [44] performs distributed calculation during the prediction time only, while our denoised bigNN method in this section distributes the calculation during both the preprocessing stage and the prediction stage (in two different ways).

**Definition 1.** *Denote $\mathcal{D}_{sub}$ as a subsample of the entire training data $\mathcal{D}$ with sample size $m$. Denote $NN(x; \mathcal{D}_{sub})$ as the nearest neighbor of $x$ among $\mathcal{D}_{sub}$. The denoised BigNN classifier is $g^\sharp(x) = g^*_{n,k,s}(NN(x; \mathcal{D}_{sub}))$. Note that this is the same as the 1-NN prediction for a pre-trained subsample in which the data points are re-labeled using the bigNN classifier $g^*_{n,k,s}$.*

We need additional assumptions to prove Theorem 3. We assume there exists some integer $d'$, named the intrinsic dimension, and constant $C_d > 0$, such that for all $x \in \mathcal{X}$ and $r > 0$, $\mu(B(x,r)) \geq C_d r^{d'}$. We will also use the VC dimension technique [42]. Although the proof makes use of some results in [44], the generalization is not trivial due to the majority voting aggregation in the bigNN classifier.

**Theorem 3.** *Let $0 < \delta < 1$. Assume VC dimension $d_{vc}$, intrinsic dimension $d'$ and constant $C_d$, the $\alpha_H$-Hölder continuity of $\eta$ and the $\beta$-margin condition, with probability at least $1 - 3\delta$ over $\mathcal{D}$,*

$$\text{Regret of } g^\sharp \leq \text{Regret of } g^*_{n,k,s} + C \left( \frac{d_{vc} \log(\frac{m}{\delta})}{mC_d} \right)^{\frac{\alpha_H(\beta+1)}{d'}}$$

Under the Hölder's condition, the regret of the bigNN classifier $g^*_{n,k,s}$ has been established to be at the rate of $N^{-\alpha_H(\beta+1)/(2\alpha_H+d)}$. Theorem 3 suggests that the pre-training step has introduced an additional error at the order of $m^{-\alpha_H(\beta+1)/d'}$, ignoring logarithmic factors. Assume for the moment that the intrinsic dimension $d'$ equals to the ambient dimension $d$ for simplicity. In this case, the additional error is relatively small compare to the original regret of bigNN, provided that the size of each subsample $m$ is at the order at least $N^{d/(2\alpha_H+d)}$. When the intrinsic dimension $d'$ is indeed smaller than the ambient dimension $d$, then the additional error is even smaller.

In principle, the subsamples at the training stage (with size $n$) and the subsamples at the prediction stage (with size $m$; from which we search for the 1NN of $x$) do not have to be the same. In practice, at the prediction stage, we may use the subsamples that are already divided up by bigNN. In other words, we do not have to conduct two different data divisions, one at the pre-training stage, the other at the prediction. In this case, to continue the discussion in the last paragraph, the additional error due to this pre-training acceleration is negligible as long as the number of total subsamples $s$ is no larger than $N^{2\alpha_H/(1\alpha_H+d)}$. Incidentally, this matches the upper bound on $s$ of $N^{2\alpha/(2\alpha+1)}$ previously.

[44] suggested to obtain multiple pre-trained 1-NN estimates from $I$ subsamples repeatedly, and conduct a majority vote among them to improve the performance. For example, they use $I = 10$ in the simulation study. The theoretical result does not depend on the number of subsamples $I$. Indeed, our proof would work even if $I = 1$. In our simulation studies, we have tried a few values of $I$ to compare the empirical performance. Our method performs fairly similarly when $I$ is greater than 9.

## 5  Simulations

All numerical studies are conducted on HPC clusters with two 12-core Intel Xeon Gold Skylake processors and four 10-core Xeon-E5 processors, with memory between 64 and 128 GB.

**Simulation 1:** We choose the split coefficient $\gamma = 0.0, 0.1 \ldots 0.9$ and $N = 1000 \times (1, 2, 3, 4, 8, 9, 16, 27, 32)$. The number of neighbors $k$ is chosen as $k_o n^{2\alpha/(2\alpha+1)} s^{-1/(2\alpha+1)}$ as stated in the theorems with $k_o = 1$, truncated at 1. The two classes are generated as $P_0 \sim N(0_5, \mathbb{I}_5)$ and $P_1 \sim N(1_5, \mathbb{I}_5)$ with the prior class probability $\pi_1 = 1/2$. The $\alpha$ value is chosen to be $\alpha_H/d = 1/5$ since the corresponding Hölder exponent $\alpha_H = 1$. In addition, the test set was independently generated with 1000 observations.

We repeat the simulation for 1000 times for each $\gamma$ and $N$. Here both the empirical risk (test error) and the empirical CIS are calculated for both the bigNN and the oracle $k$NN methods. The empirical regret is calculated as the empirical risk minus the Bayes risk, calculated using the known underlying distribution. Note that due to numerical issues and more precision needed for large $N$ and $\gamma$, the empirical risk and CIS can present some instability. The R environment is used in this study.

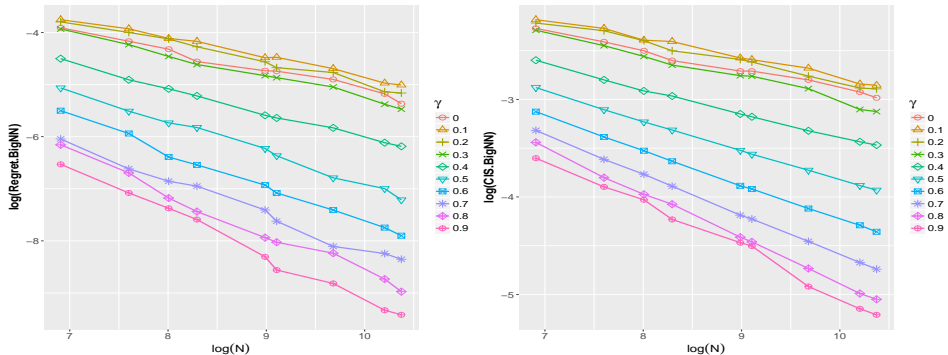

Figure 1: Regret and CIS for bigNN and oracle $k$NN ($\gamma = 0$). Different curves show different $\gamma$.

The results are reported in Figures 1. The regret and CIS lines for different $\gamma$ values are parallel to each other and are linearly decreasing as $N$ grows at the log-log scale, which verifies that the convergence rates are power functions of $N$ with negative exponents.

Inspired by the fact that the convergence rate for regret is $O(N^{-\alpha(1+\beta)/(2\alpha+1)})$ and that for CIS is $O(N^{-\alpha\beta/(2\alpha+1)})$, we fit the following two linear regression models:

$$\log(\text{Regret}) \sim \text{factor}(\gamma) + \log(N) \qquad \log(\text{CIS}) \sim \text{factor}(\gamma) + \log(N)$$

using all the dots in Figure 1. If the regression coefficients for $\log(N)$ are significant, then the convergence rates of regret and CIS are power functions of $N$, and the coefficients themselves are the exponent terms. That is, they should be approximately $-\alpha(1+\beta)/(2\alpha+1)$ and $-\alpha\beta/(2\alpha+1)$. Since the $\gamma$ term is categorical, for different $\gamma$ values, the regression lines share the common slopes, but have different intercepts. Extremely nice prediction results from these regressions are obtained. In particular, the correlation between the observed and fitted $\log$(regret) ($\log$(CIS), resp.) is 0.9916 (0.9896, resp.) The scatter plots between the observed and fitted values are shown in Figure 2, displaying almost prefect fittings. These results verify the rates obtained in our theorems.

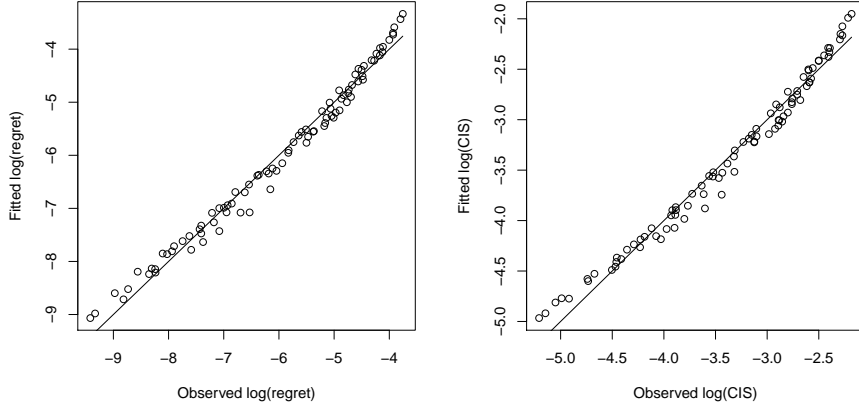

Figure 2: Scatter plots of the fitted and observed regret and CIS values.

Figure 1 in the supplementary materials shows that bigNN has significantly shorter computing time than the oracle method.

**Simulation 2**: Now suppose we intentionally fix $k$ to be a constant (this is may not be the optimal $k$). After a straightforward modification of the proofs, the rates of convergence for regret and for CIS become $O(N^{-\gamma(1+\beta)/2})$ and $O(N^{-\gamma\beta/2})$ respectively for $\gamma < 2\alpha/(2\alpha+1)$, and both regret and CIS should decrease as $\gamma$ increases. We fix number of neighbors $k = 5$, let $\gamma$ range from 0 to 0.7, and let $N = 1000 \times (1, 2, 4, 8, 10, 12, 16, 20, 32)$. The rest of the settings is the same as in Simulation 1.

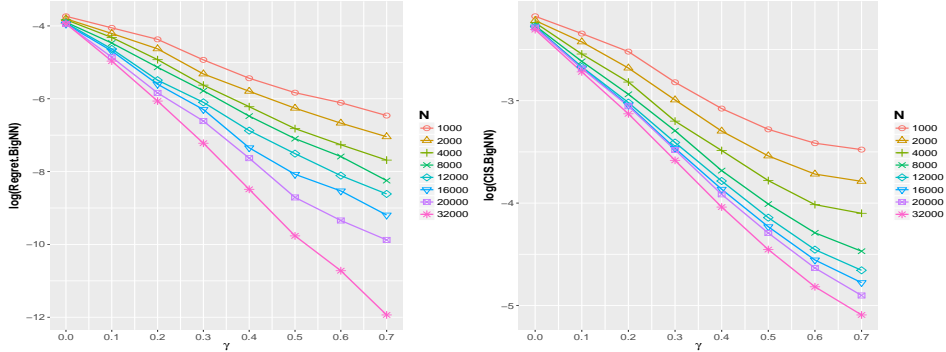

Figure 3: Regret and CIS for bigNN and oracle $k$NN ($\gamma = 0$) for $k = 5$ fixed. Different curves represent different $N$.

The results are shown in Figures 3. Both lines linearly decay in $\gamma$ (both plots are on the log scale for the $y$-axis). We note that the expected slopes in these two plots should be $-(1+\beta)/2 \times \log(N)$ and $-\beta/2 \times \log(N)$ respectively, which is verified by the figures, where larger $N$ means steeper lines.

**Simulation 3**: In Simulation 3, we compare denoised bigNN with the bigNN method. For denoised bigNN, we try to merge a few pre-training subsamples to be a prediction subsample, leading to the size of each prediction subsample $m = N^\theta$. We set $N = 27000$, $d = 8$, the pre-training split coefficient $\gamma = 0.2, 0.3$, number of prediction subsampling repeats $I = 5, 9, 13, 17, 21$, and the prediction subsample size coefficient $\theta = 0.1, 0.2, \ldots, 0.7$. The two classes are generated as

$P_1 \sim 0.5N(0_d, \mathbb{I}_d) + 0.5N(3_d, 2\mathbb{I}_d)$ and $P_0 \sim 0.5N(1.5_d, \mathbb{I}_d) + 0.5N(4.5_d, 2\mathbb{I}_d)$ with the prior class probability $\pi_1 = 1/3$. The number of neighbors $K$ in the oracle $K$NN is chosen as $K = N^{0.7}$. The number of local neighbors in bigNN are chosen as $k = \lceil k_o^* K/s \rceil$ where $k_o^* = 1.351284$, a small constant that we find works well in this example. In addition, the test set was independently generated with 1000 observations. We repeat the simulation for 300 times for each $\gamma$, $\theta$ and $I$. The results are reported in Figure 4. In each figure, the black diamond shows the prediction time and regret for the bigNN method without acceleration. Different curves represent different number of subsamples queried at the prediction stage, and their performance are similar. We see that the performance greatly changes due to the size of the subsamples at prediction $m = N^\theta$. Small $m$ (or $\theta$), corresponding to the top-left end of each curve, is fast, but introduced too much bias. Large $m$ (or $\theta$) values (bottom-right) are reasonably accurate and much faster. The computing times are shown in seconds. For this example, it seems that $\theta = 0.5$ or $0.6$ will work well.

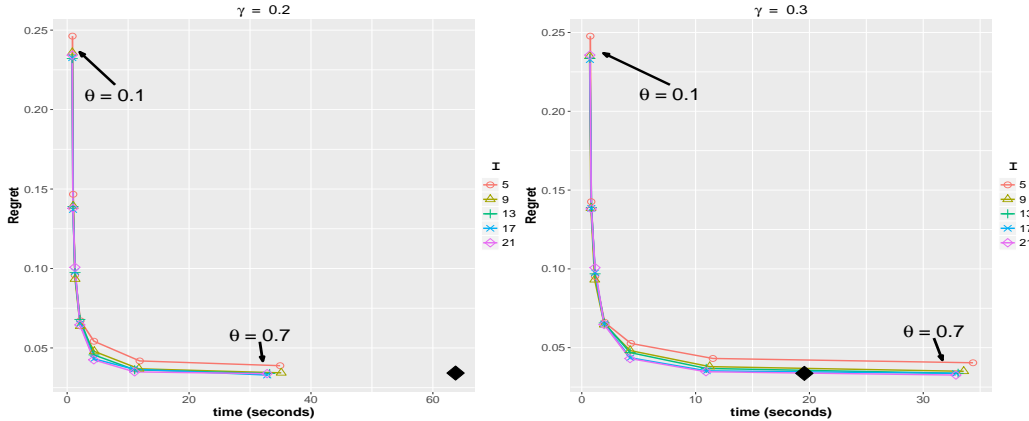

Figure 4: Regret and prediction time trade-off for denoised bigNN and bigNN (black diamonds). $\gamma = 0.2, 0.3$. $\theta = 0.1, 0.2, \ldots, 0.7$. Different curves show different $I$.

We tried the MATLAB/C++ based code from [30] on this simulation setting. To reach optimal accuracy, [30] relies on tuning two parameters knob $\alpha$ and bandwidth $h$. We tuned $h$ within $(0.1, 0.2, \ldots, 10)$ and $\alpha$ within $(1/6, 2/6, \ldots, 1)$. The best [30] classifier gave a regret 1.5 times of the denoised bigNN method. The preprocessing time (including tuning) is at the order of 4,000 seconds, compared to hundreds of seconds using our method. In terms of prediction time, the best speedup (x8) comes at the cost of the worst accuracy (2 times our regret) while the speedup with their best accuracy is only 2 to 3 folds (compared to 10 folds in our methods).

## 6 Real data examples

The OWNN method [37] gives the same optimal convergence rate of regret as oracle $k$NN and bigNN, and additionally enjoys an optimal constant factor asymptotically. A 'big'OWNN (where the base classifier is OWNN instead of $k$NN) is technically doable but is omitted in this paper for more straightforward implementations. The goal of this section is to check how much (or how little) statistical accuracy bigNN will lose even if we do not use OWNN (which is optimal in risk, both in rate and in constant) in each subset. In particular, we compare the finite-sample performance of bigNN, oracle $k$NN and oracle OWNN using real data. We note that bigNN has the same convergence rate as the other two, but with much less computing time. We deliberately choose to not include other state-of-the-art algorithms (such as SVM or random forest) in the comparison. The impact of divide and conquer for those algorithms is an interesting future research topic.

We have retained benchmark data sets HTRU2 [34], Gisette [22], Musk 1 [16], Musk 2 [17], Occupancy [8], Credit [45], and SUSY [4], from the UCI machine learning repository [33]. The test sample sizes are set as $\min(1000, \text{total sample size}/5)$. Parameters in $k$NN and OWNN are tuned using cross-validation, and the parameter $k$ in bigNN for each subsample is the optimally chosen $k$ for the oracle $k$NN divided by $s$. In Table 1, we compare the average empirical risk (test error), the empirical CIS, and the speedup of bigNN relative to oracle $k$NN, over 500 replications (OWNN typically has similar computing time as $k$NN and hence the speed comparison with OWNN is omitted). From Table 1, one can see that the three methods typically yield very similar risk and CIS

(no single method always wins), while bigNN has a computational advantage. Moreover, it seems that larger $\gamma$ values tend to have slightly worse performance for bigNN.

Table 1: BigNN compared to the oracle $k$NN and OWNN for 7 real data sets with $\gamma = 0.1, 0.2, 0.3$ except for sample data sets $N < 10000$. Speedup is defined as the computing time for oracle $k$NN divided by that for bigNN. The prefix 'R.' means risk, and 'C.' means CIS. Both are in percentage.

| DATA | SIZE | DIM | $\gamma$ | R.BigNN | R.$k$NN | R.OWNN | C.BigNN | C.$k$NN | C.OWNN | SPEEDUP |
|------|------|-----|---------|---------|---------|--------|---------|---------|--------|---------|
| HTRU2 | 17898 | 8 | 0.1 | 2.0385 | 2.1105 | 2.1188 | 0.3670 | 0.6152 | 0.5528 | 2.72 |
| HTRU2 | 17898 | 8 | 0.2 | 2.0929 | 2.1105 | 2.1188 | 0.6323 | 0.6152 | 0.5528 | 7.65 |
| HTRU2 | 17898 | 8 | 0.3 | 2.1971 | 2.1105 | 2.1188 | 0.5003 | 0.6152 | 0.5528 | 21.65 |
| GISETTE | 6000 | 5000 | 0.2 | 3.9344 | 3.5020 | 3.4749 | 4.4261 | 4.4752 | 4.3317 | 5.13 |
| MUSK1 | 476 | 166 | 0.1 | 14.7619 | 14.9767 | 14.9757 | 24.2362 | 23.0664 | 23.2707 | 1.79 |
| MUSK2 | 6598 | 166 | 0.2 | 3.8250 | 3.4400 | 3.2841 | 4.7575 | 5.1925 | 4.1615 | 5.73 |
| OCCUP | 20560 | 6 | 0.1 | 0.6207 | 0.6205 | 0.6037 | 0.3790 | 0.4431 | 0.5795 | 2.93 |
| OCCUP | 20560 | 6 | 0.2 | 0.6119 | 0.6205 | 0.6037 | 0.3717 | 0.4431 | 0.5795 | 6.97 |
| OCCUP | 20560 | 6 | 0.3 | 0.6548 | 0.6205 | 0.6037 | 0.3081 | 0.4431 | 0.5795 | 19.19 |
| CREDIT | 30000 | 24 | 0.1 | 18.8300 | 18.8681 | 18.8414 | 2.7940 | 3.5292 | 3.4392 | 3.36 |
| CREDIT | 30000 | 24 | 0.2 | 18.8467 | 18.8681 | 18.8414 | 4.3917 | 3.5292 | 3.4392 | 7.86 |
| CREDIT | 30000 | 24 | 0.3 | 18.9250 | 18.8681 | 18.8414 | 4.2496 | 3.5292 | 3.4392 | 23.22 |
| SUSY | $5000K$ | 18 | 0.1 | 19.3103 | 21.0381 | 20.7752 | 7.7034 | 7.4011 | 7.5921 | 4.59 |
| SUSY | $5000K$ | 18 | 0.2 | 21.6149 | 21.0381 | 20.7752 | 7.9073 | 7.4011 | 7.5921 | 16.76 |
| SUSY | $5000K$ | 18 | 0.3 | 22.3197 | 21.0381 | 20.7752 | 4.6716 | 7.4011 | 7.5921 | 88.22 |

In Figure 2 of the supplementary materials, we allow $\gamma$ to grow to $0.9$. As mentioned earlier, when $s$ grows too fast (e.g. $\gamma \geq 0.4$ in this example), the performance of bigNN starts to deteriorate, due to increased 'bias' of the base classifier, despite faster computing.

# 7    Conclusion

Due to computation, communication, privacy and ownership limitations, sometimes it is impossible to conduct NN classification at a central location. In this paper, we study the bigNN classifier, which distributes the computation to different locations. We show that the convergence rates of regret and CIS for bigNN are the same as the ones for the oracle NN methods, and both rates are sharp. We also show that the prediction time for bigNN can be further improved, by using the denoising acceleration technique, and it is possible to do so at a negligible loss in the statistical accuracy.

Convergence rates are only the first step to understand bigNN. The sharp rates give reassurance about worst-case behavior; however, they do not lead naturally to optimal splitting schemes or quantifications of the relative performance of two NN classifiers attaining the same rate (such as bigNN and oracle NN). Achieving these goals will be left as future works. Another future work is to prove the sharp upper bound on $\gamma$.

**Acknowledgments**

Guang Cheng's research was partially supported by the National Science Foundation (DMS-1712907, DMS-1811812, DMS-1821183,) and the Office of Naval Research (ONR N00014-18-2759). In addition, Guang Cheng is a member of the Institute for Advanced Study at Princeton University and a visiting Fellow of the Deep Learning Program at the Statistical and Applied Mathematical Sciences Institute (Fall 2019); he would like to thank both institutes for the hospitality.

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
