[Supplementary Material]

# Supplement to: Rates of Convergence for Large-scale Nearest Neighbor Classification

**Xingye Qiao**
Department of Mathematical Sciences
Binghamton University
New York, USA
qiao@math.binghamton.edu

**Jiexin Duan**
Department of Statistics
Purdue University
West Lafayette, Indiana, USA
duan32@purdue.edu

**Guang Cheng**
Department of Statistics
Purdue University
West Lafayette, Indiana, USA
chengg@purdue.edu

## 1 Notations and preliminary results

We first define some helpful notations. The support of $\mu$ is $\text{supp}(\mu) = \{x \in \mathcal{X} | \mu(B(x,r)) > 0 \text{ for all } r > 0\}$. The function $\eta$, defined for points $x \in \mathcal{X}$, can be extended to measurable set $A$ with $\mu(A) > 0$ as

$$\eta(A) = \frac{1}{\mu(A)} \int_A \eta d\mu.$$

To build a natural connection between the geometric radius and the probability measure, we define

$$r_p(x) = \inf\{r | \mu(B(x,r)) \geq p\}.$$

$r_p(x)$ is the smallest radius so that an open ball centered at $x$ has probability mass at least $p$. Intuitively, a greater $p$ leads to a greater $r_p(x)$.

We are now ready to define the so-called *effective interiors* of the two classes. The effective interior of class 1 is the set of points $x$ with $\eta(x) > 1/2$ on which the $k$-NN classifier is more likely to be correct (than on its complement):

$$\mathcal{X}_{p,\Delta}^+ = \{x \in \text{supp}(\mu) | \eta(x) > 1/2, \eta(B(x,r)) \geq 1/2 + \Delta \text{ for all } r \leq r_p(x)\}.$$

To see this, note that for a sample with $n$ points and for $r \leq r_p(x)$, because there are roughly speaking at most $np$ points in $B(x,r)$, $\eta(B(x,r)) \geq 1/2 + \Delta$ suggests that the average of the class labels of those points in $B(x,r)$ is greater than $1/2$ by at least $\Delta$; hence one can easily get a correct classification using $k$-NN at point $x$ if $p$ is $k/n$.

Similarly, the effective interior for class 0 is defined as

$$\mathcal{X}_{p,\Delta}^- = \{x \in \text{supp}(\mu) | \eta(x) < 1/2, \eta(B(x,r)) \leq 1/2 - \Delta \text{ for all } r \leq r_p(x)\},$$

and the effective boundary is defined as

$$\partial_{p,\Delta} = \mathcal{X} \backslash (\mathcal{X}_{p,\Delta}^+ \cup \mathcal{X}_{p,\Delta}^-).$$

This is the region on which the Bayes classifier and the $k$-NN are very likely to disagree.

Theorem A.1 below generalizes Theorem 5 of Chaudhuri and Dasgupta [1] to the bigNN classifier.

**Theorem A.1.** *With probability at least* $1 - \delta$,

$$\mathbb{P}_X(g^*_{n,k,s}(X) \neq g(X)) \leq \mu(\partial_{p,\Delta}) + \delta,$$

*where*

$$p = \frac{k}{n} \cdot \frac{1}{1 - \sqrt{(4/k)\log(4/\delta^{2/s})}}$$

*and*

$$\Delta = \min\left(\frac{1}{2}, \sqrt{\frac{1}{k}\log\left(\frac{4}{\delta^{2/s}}\right)}\right).$$

This theorem essentially says that the probability that the bigNN classification is different from the Bayes rule is about the size of the effective boundary which can be calibrated and controlled. The proof of the theorem starts with the evaluation of the probablity that each classifier on a local machine disagrees with the Bayes rule [1], and bounds the disagreement probability of the ensemble classifier using concentration equalities due to the Chernoff bound.

To bound the excess risk, we first consider pointwise conditional risk. The Bayes classifier has pointwise risk $R^*(x) = \min(\eta(x), 1 - \eta(x))$. The pointwise risk for the $j$th base $k$-NN classifier (the bigNN classifier, resp.) is denoted as $R^{(j)}_{n,k,s}(x)$ ($R_{n,k,s}(x)$, resp.) Next, we prove a lemma that give an upper bound for the expected pointwise regret (the expectation is with respect to the training data) under the $(\alpha, L)$-smoothness assumption of $\eta$.

**Lemma 1.** *Set $p = 2k/n$ and $\Delta_o = Lp^\alpha$. Pick any $x \in supp(\mu)$ with $\Delta(x) \equiv |\eta(x) - 1/2| > \Delta_o$. Under the $(\alpha, L)$-smoothness assumption of $\eta$,*

$$\mathbb{E}_n R_{n,k,s}(x) - R^*(x) \leq \max\left\{[8\exp(-k/8)]^{s/2},\right.$$
$$\left. 2\Delta(x)[16\exp(-2k(\Delta(x) - \Delta_o)^2)]^{s/2}\right\}$$

We are ready to prove the convergence rate of the regret for the bigNN classifier.

## 2 Proof for Theorem A.1

*Proof.* Pick any $x_o \in \mathcal{X}$ and any $0 \leq p \leq 1$, $0 \leq \Delta \leq 1/2$. Let

$$B^{(j)} = B(x_o, \rho(x_o, X^{(j)}_{(k+1)}(x_o))),$$

where $X^{(j)}_{(m)}$ is the $m$th nearest neighbor of $x_o$ in the $j$th subsample. Intuitively, $B^{(j)}$ is the ball that includes the $k$ nearest neighbors. Let $\widehat{Y}(B^{(j)})$ denote the mean of the $Y_i$'s for points $X_i \in B^{(j)}$. Then

$$\begin{aligned}
\mathbb{1}(g^{(j)}_{n,k}(x_o) \neq g(x_o)) &\leq \mathbb{1}(x_o \in \partial_{p,\Delta}) + \\
&\quad \mathbb{1}(\rho(x_o, X^{(j)}_{(k+1)}(x_o)) > r_p(x_o)) + \\
&\quad \mathbb{1}(|\widehat{Y}(B^{(j)}) - \eta(B^{(j)})| \geq \Delta).
\end{aligned} \tag{1}$$

This is the event that the $j$th base classifier does not agree with the Bayes classifier. Define the $j$th "bad event" as

$$\begin{aligned}
bad_j &= \text{BAD}_j(x_o, X^{(j)}_{1:n}, Y^{(j)}_{1:n}) \\
&= \mathbb{1}(\rho(x_o, X^{(j)}_{(k+1)}(x_o)) > r_p(x_o) \\
&\qquad \text{or } |\widehat{Y}(B^{(j)}) - \eta(B^{(j)})| \geq \Delta).
\end{aligned}$$

Now for the main event of interest here, we have

$$\mathbb{1}(g^*_{n,s,k}(x_o) \neq g(x_o)) \leq \mathbb{1}(x_o \in \partial_{p,\Delta}) + \mathbb{1}(\text{more than } \lfloor s/2 \rfloor \text{ BAD occur})$$

To see this, suppose $x_o \notin \partial_{p,\Delta}$. Then without loss of generality, $x_o$ lies in $\mathcal{X}^+_{p,\Delta}$, on which $\eta(B(x_o, r)) > 1/2 + \Delta$ for all $r < r_p(x_o)$. Next, suppose further that less than or equal to

$\lfloor s/2 \rfloor$ BAD events occurs, which means that more than $\lfloor s/2 \rfloor$ "good" events (the complements of the BAD events) occur, that is, for more than $\lfloor s/2 \rfloor$ subsets, it holds

$$\rho(x_o, X^{(j)}_{(k+1)}(x_o)) \le r_p(x_o),$$

and

$$|\widehat{Y}(B^{(j)}) - \eta(B^{(j)})| < \Delta.$$

The first inequality above means that

$$\eta(B^{(j)}) > 1/2 + \Delta.$$

These suggest that $\widehat{Y}(B^{(j)}) > 1/2$. Recall that $\eta(x_o) > 1/2$ since $x_o$ lies in $\mathcal{X}^+_{p,\Delta}$. We can conclude that on the $j$th "good" event the $j$th base $k$-NN classifier has made the same decision as the Bayes classifier. If more than half of the base $k$-NN classifiers agree with the Bayes classifier, then the bigNN classifier also agrees with the Bayes classifier.

Note that $\text{BAD}_j$'s are independent and identically distributed Bernoulli random variables. Denote $q = \mathbb{E}_N(\text{BAD}_j)$ where the expectation is taken with respect to the distribution of the training data. $\mathbb{E}_N(\text{BAD}_j)$ can be bounded using Lemmas 8 and 9 of Chaudhuri and Dasgupta [1]: for $\gamma = 1-(k/np)$

$$q = \mathbb{E}_N(\text{BAD}_j) \le \ \exp(-k\gamma^2/2) + 2\exp(-2k\Delta^2) = \delta^{4/s}/4.$$

Using concentration equalities due to the Chernoff bound [cf. Theorem 1.1, 2], we have

$$\mathbb{P}_N(\sum_{j=1}^{s} \text{BAD}_j > s/2) \le \exp(-sD(0.5\|q))$$
$$= \exp\{-s[0.5\log(0.5/q) + 0.5\log(0.5/(1-q))]\}$$
$$= \left[\frac{0.25}{q(1-q)}\right]^{-0.5s} = [4q(1-q)]^{0.5s}$$
$$\le (4q)^{0.5s} \le \delta^2$$

where $D(x\|y) = x\log(x/y) + (1-x)\log((1-x)/(1-y))$.

Taking expectation over $X_o$, we have

$$\mathbb{E}_{X_o}\mathbb{E}_N \mathbb{1}(\text{more than } \lfloor s/2 \rfloor \text{ BAD occur})$$
$$= \ \mathbb{E}_N\mathbb{E}_{X_o} \mathbb{1}(\text{more than } \lfloor s/2 \rfloor \text{ BAD occur}) \le \ \delta^2$$

Markov's inequality leads to that

$$\mathbb{P}_N(\mathbb{E}_{X_o}\mathbb{1}(\text{more than } \lfloor s/2 \rfloor \text{ BAD occur}) \ge \delta) \le \delta,$$

that is, with probability at least $1 - \delta$,

$$\mathbb{E}_{X_o}\mathbb{1}(\text{more than } \lfloor s/2 \rfloor \text{ BAD occur}) < \delta.$$

In conclusion, with probability at least $1 - \delta$,

$$\mathbb{P}_X(g^*_{n,k,s}(X) \ne g(X))$$
$$= \ \mathbb{E}_{X_o}\{\mathbb{1}(X_o \in \partial_{p,\Delta})$$
$$+ \mathbb{1}(\text{more than } \lfloor s/2 \rfloor \text{ BAD occur})\} \le \ \mu(\partial_{p,\Delta}) + \delta$$

$\square$

## 3 Proof for Lemma 1

*Proof.* Assume without loss of generality that $\eta(x) > 1/2$. The $(\alpha, L)$-smooth assumption of $\eta$ implies

$$|\eta(B(x,r)) - \eta(x)| \le L\mu(B^o(x,r))^\alpha,$$

for all $x \in \mathcal{X}$, $r > 0$. Next, for all $0 \le r \le r_p(x)$, we have

$$\eta(B(x,r)) \ge \eta(x) - Lp^\alpha = \eta(x) - \Delta_o = \frac{1}{2} + (\Delta(x) - \Delta_o).$$

Hence $x \in \mathcal{X}^+_{p,(\Delta(x)-\Delta_o)}$ and $x \notin \partial_{p,(\Delta(x)-\Delta_o)}$.

Apply the same argument as in (1), we have that

$$R^{(j)}_{n,k,s}(x) - R^*(x) = |2\eta(x) - 1| \mathbb{1}(g^{(j)}_{n,k,s}(x) \ne g(x))$$
$$= 2\Delta(x)\mathbb{1}(g^{(j)}_{n,k,s}(x) \ne g(x)) \le 2\Delta(x)\text{BAD}_j$$

where the $j$th bad event $\text{BAD}_j$ is defined as

$$\mathbb{1}[\rho(x, X^{(j)}_{(k+1)}(x)) > r_p(x) \text{ or } |\widehat{Y}(B^{(j)}) - \eta(B^{(j)})| \ge \Delta(x) - \Delta_o],$$

and $B^{(j)} = B(x, \rho(x, X^{(j)}_{(k+1)}(x)))$.

The probability of a bad event is bounded by invoking Lemma 9 and Lemma 10 in Chaudhuri and Dasgupta [1],

$$\mathbb{E}_N \text{BAD}_j \le \mathbb{P}_N(\rho(x, X^{(j)}_{(k+1)}(x)) > r_p(x)) +$$
$$\mathbb{P}_N(|\widehat{Y}(B^{(j)}) - \eta(B^{(j)})| \ge \Delta(x) - \Delta_o)$$
$$\le \exp\left(-\frac{k}{2}(1 - \frac{k}{np})^2\right) + 2\exp(-2k(\Delta(x) - \Delta_o)^2)$$
$$= \exp(-k/8) + 2\exp(-2k(\Delta(x) - \Delta_o)^2), \tag{2}$$

where we substitute $p = 2k/n$ to obtain the last equality.

Similarly, for the pointwise risk of the bigNN classifier,

$$R^*_{n,k,s}(x) - R^*(x) = 2\Delta(x)\mathbb{1}(g^*_{n,k,s}(x) \ne g(x))$$
$$= 2\Delta(x)\mathbb{1}\left(\sum_{j=1}^s \text{BAD}_j > s/2\right)$$

By taking expectation over the training data, we can then conclude that

$$\mathbb{E}_N R^*_{n,k,s}(x) - R^*(x)$$
$$\le 2\Delta(x)\mathbb{P}_N\left(\sum_{j=1}^s \text{BAD}_j > s/2\right)$$
$$\le 2\Delta(x)(4\mathbb{E}_N(\text{BAD}_j))^{s/2}$$
$$\le 2\Delta(x)[4\exp(-k/8) + 8\exp(-2k(\Delta(x) - \Delta_o)^2)]^{s/2}$$
$$\le \max\left\{[8\exp(-k/8)]^{s/2}, \right.$$
$$\left. 2\Delta(x)[16\exp(-2k(\Delta(x) - \Delta_o)^2)]^{s/2}\right\}$$

$\square$

## 4 Proof for Theorem 1

*Proof.* We define $\Delta_i = 2^i \Delta_o$. Pick any $i_o > 1$. Lemma 2 bounds the pointwise regret on the set $\Delta(x) > \Delta_{i_o}$. On the set of $\Delta(x) \le \Delta_{i_o}$, recall that

$$R^{(j)}_{n,k,s}(x) - R^*(x) = 2\Delta(x)\mathbb{1}(g^{(j)}_{n,k,s}(x) \ne g(x))$$

so that the pointwise regret is always bounded by $2\Delta(x) \le 2\Delta_{i_o}$. Then we have

$$\mathbb{E}_n R_{n,k,s} - R^*$$
$$\le \mathbb{E}_X\left\{2\Delta_{i_o} \cdot \mathbb{1}(\Delta(X) \le \Delta_{i_o}) + \right.$$
$$\max\left\{[8\exp(-k/8)]^{s/2}, \right.$$
$$\left. 2\Delta(x)[16\exp(-2k(\Delta(x) - \Delta_o)^2)]^{s/2}\right\} \cdot \mathbb{1}(\Delta(X) > \Delta_{i_o})\right\}$$

For the first term,
$$\mathbb{E}_X\{2\Delta_{i_o}\mathbb{1}(\Delta(X) \le \Delta_{i_o})\} \le 2C\Delta_{i_o}^{\beta+1}.$$

For the second term,
$$\mathbb{E}_X\left\{2[4e^{-k/8}]^{s/2}\mathbb{1}(\Delta(X) > \Delta_{i_o})\right\} \le (2\sqrt{2})^s \exp(-ks/16).$$

The last term is decomposed to the sum of the followings with $i \ge i_o$:
$$\mathbb{E}_X\{2\Delta(x)[16\exp(-2k(\Delta(x) - \Delta_o)^2)]^{s/2}$$
$$\times \mathbb{1}(\Delta_i < \Delta(X) \le \Delta_{i+1})\}$$
$$\le 2\Delta_{i+1}[16\exp(-2k(\Delta_i - \Delta_o)^2)]^{s/2}\mathbb{P}_X(\Delta(X) \le \Delta_{i+1})$$
$$\le 2C\Delta_{i+1}^{1+\beta}4^s\exp(-ks(\Delta_i - \Delta_o)^2).$$

If we set $i_o = \max\left(1, \left\lceil \log_2 \sqrt{\frac{2(2+\beta)}{ks\Delta_o^2}} \right\rceil\right)$,
$$\frac{2C\Delta_{i+1}^{1+\beta}4^s\exp(-ks(\Delta_i - \Delta_o)^2)}{2C\Delta_i^{1+\beta}4^s\exp(-ks(\Delta_{i-1} - \Delta_o)^2)}$$
$$= 2^{1+\beta}\exp(-ks[(\Delta_i - \Delta_o)^2 - (\Delta_{i-1} - \Delta_o)^2])$$
$$= 2^{1+\beta}\exp(-ks\Delta_o^2[(2^i - 1)^2 - (2^{i-1} - 1)^2])$$
$$\le 2^{1+\beta}\exp(-ks\Delta_o^2[2^{i-1} \cdot 2 \cdot 2^{i-1}])$$
$$\le 2^{1+\beta}\exp(-(2+\beta)) \le 1/2.$$

Therefore,
$$\mathbb{E}_X\{2\Delta(x)[16\exp(-2k(\Delta(x) - \Delta_o)^2)]^{s/2}$$
$$* \mathbb{1}(\Delta(X) > \Delta_{i_o})\}$$
$$\le \sum_{i=i_o}^{\infty}\mathbb{E}_X\{2\Delta(x)[16\exp(-2k(\Delta(x) - \Delta_o)^2)]^{s/2}$$
$$* \mathbb{1}(\Delta_i < \Delta(X) \le \Delta_{i+1})\}$$
$$\le 2\mathbb{E}_X\{2\Delta(x)[16\exp(-2k(\Delta(x) - \Delta_o)^2)]^{s/2}$$
$$\mathbb{1}(\Delta_{i_o} < \Delta(X) \le \Delta_{i_o+1})\}$$
$$\le 4C\Delta_{i_o+1}^{1+\beta}4^s\exp(-ks(\Delta_{i_o} - \Delta_o)^2) \le C_1\Delta_{i_o}^{1+\beta}$$

Hence,
$$\mathbb{E}_n R_{n,k,s} - R^*$$
$$\le 2C\Delta_{i_o}^{1+\beta} + (2\sqrt{2})^s\exp(-ks/16) + C_1\Delta_{i_o}^{1+\beta}$$
$$\le 2C_2\Delta_{i_o}^{1+\beta} = 2C_2 \cdot 2^{i_o(1+\beta)}\Delta_o^{1+\beta}$$
$$= 2C_2 \cdot \left\{\max\left[2\Delta_o, \sqrt{\frac{2(2+\beta)}{ks}}\right]\right\}^{1+\beta}$$
$$= 2C_2 \cdot \left\{\max\left[2L(2k/n)^{\alpha}, \sqrt{\frac{2(2+\beta)}{ks}}\right]\right\}^{1+\beta}.$$

Recall that $s = N^\gamma$ and $n = N^{1-\gamma}$. Therefore, if we let $k = k_o n^{2\alpha/(2\alpha+1)}s^{-1/(2\alpha+1)}$, then
$$\mathbb{E}_n R_{n,k,s} - R^*$$
$$\le C_0[n^{-\alpha/(2\alpha+1)}s^{-\alpha/(2\alpha+1)}]^{1+\beta}$$
$$= C_0 N^{-\alpha(1+\beta)/(2\alpha+1)}$$

$\square$

# 5  Proof for Theorem 2

*Proof.* Assume that $\Delta(x) > \Delta_o$. Assume without loss of generality that $\eta(x) > 1/2$. Under the $(\alpha, L)$-smooth assumption of $\eta$, we have that

$$\eta(B(x, r)) \geq \eta(x) - Lp^\alpha$$
$$= \eta(x) - \Delta_o = \frac{1}{2} + (\Delta(x) - \Delta_o).$$

Hence $x \in \mathcal{X}^+_{p,((\Delta(x)-\Delta_o))}$ and $x \notin \partial_{p,(\Delta(x)-\Delta_o)}$.

Let $g^*_{n,s,k}(x_o)$, $\tilde{g}_{n,s,k}(x_o)$ be the bigNN classifiers based on two iid samples respectively. Apply the same argument as in (1), it is easy to see that given $x \notin \partial_{p,(\Delta(x)-\Delta_o)}$,

$$\mathbb{1}(g^*_{n,s,k}(x_o) \neq \tilde{g}_{n,s,k}(x_o))$$
$$= \mathbb{1}(\text{more than } \lfloor s/2 \rfloor \text{ BAD and fewer than } \lfloor s/2 \rfloor \text{ BAD}', \text{ or}$$
$$\text{fewer than } \lfloor s/2 \rfloor \text{ BAD and more than } \lfloor s/2 \rfloor \text{ BAD}')$$

Taking expectation over the training data, we have

$$CIS(\text{bigNN})(x) \equiv \mathbb{P}_N(g^*_{n,s,k}(x_o) \neq \tilde{g}_{n,s,k}(x_o))$$
$$\leq 2\mathbb{P}_N \left( \sum_{j=1}^s \text{BAD}_j > s/2 \text{ and } \sum_{j=1}^s \text{BAD}'_j < s/2 \right)$$
$$\leq 2\mathbb{P}_N \left( \sum_{j=1}^s \text{BAD}_j > s/2 \right)$$

$$\leq 2[4\exp(-k/8) + 8\exp(-2k(\Delta(x) - \Delta_o)^2)]^{s/2}$$
$$\leq 4\max \left\{ [4\exp(-k/8)]^{s/2}, \right.$$
$$\left. [8\exp(-2k(\Delta(x) - \Delta_o)^2)]^{s/2} \right\}$$

The rest of the proof follows that of Theorem 3. Observe that the optimal rate of convergence is $O(N^{-\alpha\beta/(2\alpha+1)})$. $\qquad\square$

# 6  Proof for Theorem 3

*Proof.* For simplicity, denote $\eta_1 \triangleq \eta$ and $\eta_0 \triangleq 1 - \eta$. Let $x' \triangleq \text{NN}(x; \mathcal{D}_{sub})$. We consider the event $A(x) \triangleq \mathbb{1}\{|\eta(x) - 1/2| \geq \phi\}$ where $\phi \triangleq C \left( \frac{d_{vc} \log(\frac{m}{\delta})}{mC_d} \right)^{\frac{\alpha_H}{d'}}$ and its complement $\bar{A}(x)$.

On the set $A(x)$, it follows that $\eta_g(x) - \eta_{1-g}(x) \geq 2\phi$ due to the definition of $A(x)$. Consider the inequality

$$\eta_g(x) - \eta_{g^\sharp}(x) = \eta_g(x) - \eta_{g^*_{n,k,s}}(x') \leq |\eta_g(x) - \eta_g(x')| + [\eta_g(x') - \eta_{g^*_{n,k,s}}(x')] \qquad (3)$$

The first term on the right hand side of (3) is

$$|\eta_g(x) - \eta_g(x')| \leq |\eta(x) - \eta(x')| \leq L\|x - x'\|^{\alpha_H} \leq C \left( \frac{d_{vc} \log(\frac{m}{\delta})}{mC_d} \right)^{\frac{\alpha_H}{d'}} = \phi$$

with probability $1 - \delta$ due to the Hölder-smoothness of $\eta$ and Lemma 1 of Xue and Kpotufe [3].

Moreover, define the $j$th bad event as $\text{BAD}_j \triangleq \mathbb{1}\{\hat{\eta}_j(x') - \eta(x') > \phi\}$ for $j = 1, \ldots, s$. On the set $A(x)$, $\eta_g(x') - \eta_{g^*_{n,k,s}}(x') \neq 0$ if and only if $g(x') \neq g^*_{n,k,s}(x')$. The latter event implies that

more than $s/2$ bad events occur, whose probability is less than $(4\mathbb{E}(\mathrm{BAD}_j))^{s/2}$ using concentration equalities due to the Chernoff bound. Let $\tilde{\phi} \triangleq C\left(\frac{d_{vc}\log(2n/\tilde{\delta})}{nC_d}\right)^{\frac{\alpha_H}{2\alpha_H+d'}}$. Choose $\tilde{\delta}$ so that $\tilde{\phi} < \phi$ From Proposition 1 of Xue and Kpotufe [3], we know that

$$\mathbb{E}(\mathrm{BAD}_j) = \mathbb{P}[\hat{Y}^{(j)}(x') - \eta(x') > \phi] < \mathbb{P}\left[\hat{\eta}_j(x') - \eta(x') > \tilde{\phi}\right] \leq 2\tilde{\delta}.$$

Therefore, with probability at least $1 - (8\tilde{\delta})^{s/2}$, the second term $\eta_g(x') - \eta_{g^*_{n,k,s}}(x') = 0$. For large enough $s$ and $n$, $(8\tilde{\delta})^{s/2} < \delta$. Hence the above equality occurs with probability at least $1 - \delta$.

Combining the results, we have with probability at least $1 - 2\delta$,

$$\eta_g(x) - \eta_{g^\sharp}(x) < 2\phi \leq \eta_g(x) - \eta_{1-g}(x)$$

so that $\eta_{g^\sharp}(x) > \eta_{1-g}(x)$, which means $g^\sharp(x) = g(x)$. In this case, the excess error is 0.

We now consider the set of $\bar{A}(x) \triangleq \mathbb{1}\{|\eta(x) - 1/2| < \phi\}$. Consider the following inequality,

$$\eta_g(x) - \eta_{g^\sharp}(x) = \eta_g(x) - \eta_{g^*_{n,k,s}}(x') \leq [\eta_g(x) - \eta_{g^*_{n,k,s}}(x)] + |\eta_{g^*_{n,k,s}}(x) - \eta_{g^*_{n,k,s}}(x')| \quad (4)$$

The first term on the right hand side will be related to the regret of the BigNN classifier. For the second term, by Lemma 1 of Xue and Kpotufe [3], we have with probability at least $1 - \delta$ over the sample,

$$|\eta(x) - \eta(x')| \leq L\|x - x'\|^\alpha \leq \phi.$$

Then $|\eta(x') - 1/2| \leq 2\phi$. We have

$$|\eta_{g^*_{n,k,s}}(x) - \eta_{g^*_{n,k,s}}(x')| \leq \sup_{\substack{|\eta(x)-1/2|<\phi \\ |\eta(x')-1/2|<2\phi}} |\eta(x) - \eta(x')| < 3\phi.$$

Lastly, take expectation over $X$ of the left hand side of both (3) and (4), we have with probability at least $1 - 3\delta$,

$$\begin{aligned}
\text{Regret of } g^\sharp &= \mathbb{E}_X(\eta_g(X) - \eta_{g^\sharp}(X)) \\
&= \mathbb{E}_X[0A(X)] + \mathbb{E}_X(\eta_g(X) - \eta_{g^\sharp}(X))\bar{A}(X) \\
&\leq \mathbb{E}_X[\eta_g(X) - \eta_{g^*_{n,k,s}}(X)] + \mathbb{E}_X|\eta_{g^*_{n,k,s}}(X) - \eta_{g^*_{n,k,s}}(X')|\bar{A}(X) \\
&\leq \text{Regret of } g^*_{n,k,s} + 3\phi\mathbb{E}_X\bar{A}(X) \\
&= \text{Regret of } g^*_{n,k,s} + 3\phi\mathbb{P}(|\eta(x) - 1/2| < \phi) \\
&\leq \text{Regret of } g^*_{n,k,s} + C(\phi)^{\beta+1} \\
&\leq \text{Regret of } g^*_{n,k,s} + C\left(\frac{d_{vc}\log(\frac{m}{\delta})}{mC_d}\right)^{\frac{\alpha_H(\beta+1)}{d'}}.
\end{aligned}$$

$\square$

## 7 Additional figures

Figure 1 shows that bigNN has significantly shorter computing time than the oracle method. Here the 'speedup' is defined as the computing time for the oracle $k$NN divided by the time for bigNN.

In Figure 2, we allow $\gamma$ to grow to 0.9. As mentioned earlier, when $s$ grows too fast (e.g. $\gamma \geq 0.4$ in this example), the performance of bigNN starts to deteriorate, due to increased 'bias' of the base classifier, despite faster computing.

Figure 1: Speedup of bigNN, for $\gamma = 0.0, 0.1, \ldots, 0.9$, with $k$ set as $n^{2\alpha/(2\alpha+1)}s^{-1/(2\alpha+1)}$. $\gamma = 0$ corresponds to the oracle $k$NN.

Figure 2: Risk and speedup of bigNN, $\gamma = 0.0, \ldots, 0.9$, for the Credit data.