[Reviews · NeurIPS 2019]

Reviewer 1



The paper presents a simple and clear algorithm of a divide-and-conquer scheme of distributed classification using the nearest neighbour framework. The general methods presented are not entirely new, but are accompanied by a crisp statistical analysis which proves tight convergence rates to the Bayes optimal classifier. The novelty in this algorithm is the complete distributed nature of it - the fact that very little information must be transferred between different computing units as opposed to previous work algorithms which compelled more data transference between them. The claims are clear and understandable, and the theoretical part is very satisfactory, as well as a nice complementary empirical section showing improved speeds (in the denoised case) as well as improved regret. On the other hand, I add a few points to explain the overall score I have given this submission: 1) The algorithm is rather similar to the denoising algorithm referred to in the paper. The denoising algorithm could be performed in a distributed manner as well since it performs calculations on different subsamples of the data. In addition, similar analysis to that of the submitted paper has been done in the denoising case. 2) In the extreme cases in which the subsamples are very small (the size of s is bounded from above but still may reach significantly large sizes) it seems there may be a possibility that k >1/|S|. I may be mistaken regarding this statement, but if it is true than a tighter higher bound is necessary. 3) The idea of distributing the data and thus speed up the process of classification seems somewhat inferior to compressing the data. the speedup occurs from multiprocessing rather than reduction of computations (There is still need to entirely search each subsample). The denoising variation of the algorithm presents actual reduction in computation time, which the original BigNN does not. Therefore, it seems to me that a fair amount of work presented here is due to the referenced papers. Finally, I would like to state that the paper is well written and is very clean and understandable. The only downside in my eye is the fact a lot of its originality comes from previous papers. With further analysis answering the questions presented in the conclusions this could be a more novel and unique paper. Best of luck

Reviewer 2



After reading the author feedback, I am impressed by the thorough comments and I am increasing my score from 5 to 7. Please incorporate your discussion points to the paper, and thanks for the well-thought-out response! * * * * * Original review below * * * * * From a methodological standpoint, this paper combines an ensembling approach that is basically just pasting (Brieman 1999) with nearest neighbor classification and then also with the denoising method of Xue and Kpotufe (2018). We already know that k-NN classification can achieve the minimax optimal rate in fairly general settings (Chaudhuri and Dasgupta 2014). That pasting used in conjunction with k-NN classification also achieves this rate is unsurprising (and the analysis is straightforward), and stitching the result with the theory developed by Xue and Kpotufe is also straightforward. Overall the theoretical contributions are reassuring albeit unsurprising and incremental. Given the aim of the paper, I think it's extremely important to experimentally compare against recently proposed quantization schemes that are largely about how to scale up nearest neighbors to large datasets: see Kpotufe and Verma 2017, and Kontorovich, Weiss, and Sabato 2017. Perhaps more discussion/theory on when/why one should use pasting rather than bagging to ensemble k-NN estimators would be helpful. I realize that the authors have mentioned a little bit about known asymptotic results but I think numerical simulations would really be helpful here. Separately, depending on the dataset (i.e., the feature space and distribution), I would suspect that even just taking 1 subsample without ensembling could yield a good classifier. Perhaps some discussion on understanding how much training data we could get away with (and whether we could just ignore a lot of the data to save on computation) could be helpful. This comment is a bit related to the quantization strategies mentioned above.

Reviewer 3



This paper treats the theoretical analysis of k-nearest neighbor (kNN) classification when the kNN classification is performed separately in different machines with different data. The information is combined later for final decision. The paper is clearly written and the experiments support the derived theory well. The derived result is important because the simple methods such as k-NN classification is becoming important for treating really large data, but the data now cannot be treated in a single machine. The results achieve the similar order of convergence rate in a single machine algorithm without loosing the simplicity of k-NN methods. This paper is a nice contribution to nearest neighbor community in NeurIPS.

Reviewer 4



The paper is well-written, the results appear to be novel and correct.

[Author Response · NeurIPS 2019]

**Reviewer 1:** *"The algorithm is rather similar to the denoising algorithm referred to in the paper. The denoising algorithm could be performed in a distributed manner as well since it performs calculations on different subsamples of the data."* The second part of our paper (denoised bigNN) may be viewed as an improvement of the denoising algorithm since it not only leverages the denoising technique to shorten the prediction time, but also shortens the preprocessing time by distributed learning. Indeed, the two algorithms share some similarity. However, a subtle difference lies in the fact that the denoising algorithm performs distributed calculation during the prediction time only, while our denoised bigNN distributes the calculation during both the preprocessing time and the prediction time.

*"In the extreme cases... it seems there may be a possibility that $k > 1/|S|$."* Because $1/|S|$ is a fraction, it is always true that $k > 1/|S|$. You were probably concerned that $k > N/s$, i.e., $k$ could be greater than the subsample size. We would like to clarify that this would never happen since our choice $k = k_o n^{2\alpha/(2\alpha+1)} s^{-1/(2\alpha+1)}$ automatically satisfies $k < n = N/s$ for a fixed $k_o$ and large enough $N$.

*"The idea of distributing the data and thus speed up the process of classification seems somewhat inferior to compressing the data. the speedup occurs from multiprocessing rather than reduction of computations (There is still need to entirely search each subsample). The denoising variation of the algorithm presents actual reduction in computation time, which the original BigNN does not..."* It depends on how one defines computation time. If one also takes into account the preprocessing time, denoising (or data compression) merely shifts most of the computation time from the prediction stage to the preprocessing stage; but the computation still has to be done. One could argue that the prediction time is what really matters. However, without the likes of the proposed distributed algorithm, preprocessing will be very difficult due to the very large data volume. Our proposed denoised bigNN algorithm not only shortens the preprocessing time of the original denoising algorithm, but also retains the optimal accuracy. We thank you for the other positive comments and will address the relation between the proposed method and the denoising method more clearly in the camera-ready version, if accepted.

**Reviewer 2:** *"Given the aim of the paper, I think it's extremely important to experimentally compare against recently proposed quantization schemes that are largely about how to scale up nearest neighbors to large datasets: see KV17, and KWS17."* KWS17 is theory oriented and the authors did not provide code. We tried the MATLAB/C++ based code for the KV17 method on Simulation 3. To reach optimal accuracy, KV17 relies on tuning two parameters knob $\alpha$ and bandwidth $h$. We tuned $h$ within $(0.1, 0.2, \ldots, 10)$ and $\alpha$ within $(1/6, 2/6, \ldots, 1)$. The best KV17 classifier gave a regret 1.5 times of the denoised bigNN method. The preprocessing time (including tuning) is at the order of 4,000 seconds, compared to hundred seconds using our method. In terms of prediction time, the best speedup (x8) comes at the cost of the worst accuracy (2 times our regret) while the speedup with their best accuracy is only 2 to 3 folds (compared to 10 folds in our methods). Note our R-based code (to be released publicly) has room for improvement. Philosophically quantization schemes share similarity with the denoising scheme: both referred works start with a $r$-net, which is a collection of data points that quantize the training data. They correspond to cells of a Voronoi partition of the entire sample space. The average $Y$ value or the majority class of those training points that fall into each cell is then assigned to these cells. This is quite similar to the denoising scheme in Xue and Kpotufe (2018), in which quantization is achieved by random subsamplings, and pre-labelling the points in the subsample in the denoising algorithm works like "averaging the $Y$ values" in the quantization scheme. Under the hood these quantization schemes still have heavy computational burden in terms of preprocessing: for a very large training data, assigning the weights for each cell will be as difficult as predicting the class label of a query point using $k$NN. From this perspective, our denoised bigNN algorithm has the potential to improve quantization methods by shortening the preprocessing time without scarificing the accuracy (in this paper we have shown the case for the denoising algorithm, a special kind of quantization method).

*"Perhaps more discussion/theory on when/why one should use pasting rather than bagging to ensemble k-NN estimators would be helpful..."* Indeed both pasting and bagging can ensemble estimators. Our algorithm is motivated by the need to maintain data decentralisation/privacy and enhance speed performance, while bagging (or bootstrap in general), historically, was proposed to (1) enhance the prediction accuracy by reducing variance and (2) conduct valid statistical inference even when the sample size is not large enough. Neither is our concern here since the sample size is not too small but too large, and by proving the convergence rate we show that the optimal prediction accuracy is retained.

*"Separately, depending on the dataset (i.e., the feature space and distribution), I would suspect that even just taking 1 subsample without ensembling could yield a good classifier. Perhaps some discussion on understanding how much training data we could get away with (and whether we could just ignore a lot of the data to save on computation) could be helpful..."* Indeed, even Xue and Kpotufe (2018) suggested that a small number of subsamples (repitions) would suffice. Our proof would work even for only one subsample. However, we must point out that denoising and quantization does not work by ignoring a lot of data all together, but by extraditing the information ahead of time (during preprocessing) and not bothering with the entire data later on. As far as the question of how sparse the quantization can be, we had a relevant comment in line 221 which suggested that the smallest each subsample can be is $N^{1/(2\alpha+1)}$.

**Reviewer 3 and Reviewer 4:** We thank you for your very positive comments.

[Meta-Review · NeurIPS 2019]

The referees are unanimous in recommending acceptance.